# Why Equivariant Networks Lose Information: Invariant Rings and the Role of Aggregation

## Abstract

Equivariant neural networks exhibit well-documented expressivity limitations: rotation-equivariant networks collapse directional information to radial features, and matrix-equivariant networks show rank degeneracy. We provide an algebraic account of these phenomena using two complementary frameworks: classical (compact-group) invariant theory and Sato–Kimura prehomogeneous vector space (PVS) theory, which handles non-compact actions admitting an open dense orbit. This allows us to organize a body of recent results—encompassing geometric GNN completeness, body-order, sparse-graph rigidity, stabilizer obstructions, and tensor-product selection rules—under a common algebraic frame. Escape from these constraints comes from aggregation: the transition from a single fiber $V$ to the product $V^n$ enriches the invariant ring with cross-invariants, and this transition rather than network depth is what enables discrimination of geometric relationships. We use this frame to analyze modern architectures—SchNet, PaiNN, DimeNet, MACE, HEGNN, GotenNet—through their body-order, the largest $n$ for which they access the invariant ring of $V^n$. Illustrative experiments verify the predicted SO(3)-vs-O(3) and aggregation-vs-fiber-only gaps.

## 1 Introduction

### 1.1 Equivariant Neural Networks and Their Importance

Many problems in science and engineering involve data with inherent symmetries. Molecules look the same regardless of how we orient them in space; the laws of physics do not depend on our choice of coordinate system; and the properties of a crystal are unchanged by the symmetries of its lattice. When building neural networks to learn from such data, it is natural to ask that the network respect these symmetries: if the input is rotated, the output should transform accordingly.

This requirement defines equivariant neural networks. Formally, given a group $G$ acting on input space $V$ and output space $W$, a function $\Phi : V \to W$ is $G$-equivariant if $\Phi(g \cdot v) = g \cdot \Phi(v)$ for all group elements $g$ and inputs $v$. When the output is a scalar (unchanged by the group action), the function is called $G$-invariant.

Equivariant architectures have proven highly successful in practice. In computational chemistry and materials science, machine-learned interatomic potentials based on rotation-equivariant networks (Thomas et al., 2018; Schütt et al., 2018; Batzner et al., 2022) can predict molecular properties with near quantum-mechanical accuracy at a fraction of the computational cost. These potentials now enable molecular dynamics simulations that were previously intractable, with applications ranging from drug discovery to catalyst design. In computer vision, translation-equivariant convolutional networks (Cohen and Welling, 2016) remain the foundation of image recognition. In physics simulation, equivariant graph networks (Sanchez-Gonzalez et al., 2020) learn to predict the dynamics of complex systems. By building symmetry into the architecture, we reduce the hypothesis space, improve sample efficiency, and guarantee physically sensible predictions.

## 1.2 The Problem: Expressivity Limitations

Equivariance constraints impose fundamental limitations on what functions a network can represent. These limitations take several forms:

Scalar collapse. Rotation-equivariant networks processing single 3D vectors are algebraically constrained to collapse directional information to radial features. An SO(3)-equivariant map $\Phi : \mathbb{R}^3 \to \mathbb{R}^3$ must have the form $\Phi(v) = p(|v|^2) \cdot v$ (Weyl, 1946; Blum-Smith and Villar, 2023), where $p$ is a scalar function of the squared norm. The direction of $v$ is preserved but cannot be discriminated—two vectors pointing in different directions but with the same length produce proportional outputs.

Rank collapse. Networks equivariant to matrix transformations exhibit degeneracy at low rank. When $\mathrm{GL}(n) \times \mathrm{GL}(n)$ acts on $n \times n$ matrices by $(A, B) \cdot M = AMB^{-1}$, the action is transitive on invertible matrices. This is a non-compact action that lies outside the reach of classical Weyl-style invariant theory; the natural framework is Sato–Kimura prehomogeneous vector space (PVS) theory. The polynomial invariant ring is just constants ($k[\mathrm{Mat}_n]^G = k$), so invariant scalar features carry no information. The geometry is governed by the relative invariant $\det(M)$, which defines the singular set, and the rank stratification of that set.

Orbit limitations. More generally, equivariant networks cannot distinguish inputs that lie on the same group orbit. This is a fundamental constraint, but the practical severity depends on the orbit structure: when most inputs lie in a single large orbit, the network loses discriminating power.

These constraints have motivated theoretical analyses of equivariant network expressivity (Joshi et al., 2023; Pacini et al., 2024; Villar et al., 2021) and various architectural innovations to circumvent them. The question we address is: why do these limitations arise algebraically, and what determines their severity?

## 1.3 The Knowledge Gap

Two largely independent lines of research address equivariant network expressivity, but neither provides an algebraic explanation of why certain representations lead to expressivity limitations.

The machine learning community has developed sophisticated tools for analyzing what networks can do. The Weisfeiler-Leman hierarchy (Xu et al., 2019; Morris et al., 2019) characterizes the discriminating power of graph neural networks. Joshi et al. (2023) extend this to geometric graphs with their Geometric Weisfeiler-Leman (GWL) test, which assigns colors to geometric neighborhoods via an idealized "I-HASH" function assumed to be orbit-injective; GWL determines what geometric GNNs can and cannot distinguish—for instance, that invariant layers cannot distinguish 1-hop identical graphs. For the geometric WL hierarchy itself, Delle Rose et al. (2023) establish a concrete completeness threshold: $(d-1)$-WL distinguishes any two non-isometric point clouds in $\mathbb{R}^d$ after three iterations, with 2-WL complete in $\mathbb{R}^3$ in particular. Sverdlov and Dym (2025) extend this picture to the sparse-graph regime more typical of practice, showing that generic completeness depends on whether intermediate features are scalars (requiring graph rigidity) or vectors (requiring only connectivity). Li et al. (2023) give a parallel constructive picture specialized to the distance-matrix setting, introducing $k$-DisGNNs (Distance GNNs) that process $k$-tuples of nodes via $k$-WL or its Folklore variant $k$-FWL on the all-pair distance graph; they establish scalar universality at $k \geq 2$ and vector universality at $k \geq 3$, and prove that DimeNet and GemNet arise as special cases of 2- and 3-DisGNNs respectively. Universality results address a different question: Dym and Maron (2020) establishes conditions under which rotation-equivariant point cloud networks can approximate any equivariant function; Villar et al. (2021) showed that scalar features suffice for universal approximation; Kondor (2025) provides a comprehensive treatment of equivariant architectures for physics. More recently, Cen et al. (2025) provide a constructive complement, decomposing complete equivariant networks into a complete graph-level scalar invariant (a "canonical form" $\Gamma(\mathcal{G})$) and a full-rank steerable basis set, and exhibiting a polynomial-time canonical form for asymmetric geometric graphs via four-point positioning. These results answer whether networks can represent certain function classes, or how to construct one explicit complete invariant in favorable cases, but do not explain why particular representations impose particular constraints. In the GWL framework, I-HASH is treated as a black box—the theory does not specify which invariants it actually computes.

Classical invariant theory, recently made accessible to machine learning by Blum-Smith and Villar (2023), provides tools for parameterizing equivariant maps. Given a group action, Malgrange's method constructs explicit bases for equivariant polynomial maps in terms of generating invariants (Weyl, 1946; Kraft and Procesi, 2000). Puny et al. (2023) enumerated equivariant polynomial bases for specific groups. This work answers how to build equivariant networks, but the connection between invariant ring structure and practical expressivity limitations—why some representations are severely constrained while others are not—has not been systematically developed.

The gap we address is the question of constraints and impossibilities: which invariants are available to a network, which are not, and what algebraic structure explains the difference? For the GWL framework, we provide the algebraic content of I-HASH: the First Fundamental Theorem specifies exactly which invariants are computable (e.g., norms and dot products for O(3)), explaining why the discrimination limits that Joshi et al. identify arise. The constructive framework of Cen et al. (2025) is dual: rather than characterizing the ring of invariants, they build one explicit complete invariant, with the symmetric case (configurations with nontrivial geometric stabilizer) left open. Our algebraic perspective identifies why that case is structurally hard—it is precisely where equivariant outputs are confined to a proper subspace determined by the stabilizer's representation. More broadly, prehomogeneous vector space theory (Sato and Kimura, 1977; Kimura, 2003), which classifies representations with open orbits and characterizes their relative invariants, provides precisely the tools needed for non-compact actions of this kind, where compact-group FFTs do not extend. To our knowledge, this paper is the first to identify PVS theory as a natural framework for analyzing equivariance constraints on non-compact actions with open orbits in machine learning.

### 1.4  Our Approach and Contributions

We analyze expressivity limitations through two classical mathematical theories:

Classical invariant theory, particularly the First Fundamental Theorems (FFTs) that characterize the generators of invariant rings for classical groups. For SO(3) acting on $\mathbb{R}^3$, the FFT states that all invariants are generated by the squared norm $|v|^2$, and all equivariant maps $\mathbb{R}^3 \to \mathbb{R}^3$ are generated by the identity map $v \mapsto v$ with coefficients in this invariant ring. This immediately explains scalar collapse.

Prehomogeneous vector space (PVS) theory, developed by Sato and Kimura (1977), which classifies representations $(G, V)$ where the group has an open dense orbit. For such representations, there exists a unique (up to character) relative invariant whose zero set is the complement of the open orbit. When $(G, V)$ is a PVS, equivariant maps on the generic orbit are severely constrained.

A key contribution is clarifying when each framework applies. Classical FFT-style invariant theory addresses compact-group actions: SO(3) on $\mathbb{R}^3$ has a finitely generated invariant ring ($\mathbb{R}[|v|^2]$), and scalar collapse follows from the FFT. PVS theory is the natural framework when the action is non-compact and admits an open dense orbit: $(\mathrm{GL}(n) \times \mathrm{GL}(n), \mathrm{Mat}_n)$ is the standard worked example from Sato–Kimura's classification, with a single open orbit, a trivial polynomial invariant ring, and all algebraic content sitting in the relative invariant $\det(M)$. The two frameworks describe distinct algebraic regimes that are commonly conflated under the slogan "invariant networks lose information."

Expressivity limitations of equivariant networks are empirically known. Our contribution is identifying which mathematical framework governs each case. The two regimes have sharply different algebraic content: compact-group invariant theory gives non-trivial invariant rings with codimension-1 orbits (e.g., spheres for SO(3)), while PVS theory gives trivial polynomial invariant rings with single open orbits and relative invariants picking up the algebraic content. Conflating them obscures both the source of each constraint and the routes for escaping it.

Within this account, the constraints have a clean escape route. We show that aggregation (the move from a single fiber $V$ to product representations $V^n$) enriches the invariant ring and enables discrimination of geometric relationships that single-fiber processing cannot access. For SO(3) on $(\mathbb{R}^3)^n$, the invariant ring is generated by all pairwise dot products and norms, providing access to angular information. This explains why message-passing architectures, which aggregate information from multiple neighbors, succeed where per-edge processing fails. The resulting body-order viewpoint (the largest $n$ for which an architecture accesses $k[V^n]^G$)

then organizes existing architectures (SchNet, PaiNN, DimeNet, MACE, HEGNN, GotenNet) under a single algebraic parameter, and recovers a number of recent completeness, sparse-graph, and chirality results as consequences of the same algebraic structure.

We also clarify the role of depth. Contrary to what one might hope, increasing network depth does not expand the class of representable functions for single-fiber inputs—it only improves approximation quality within that class. Depth helps approximate complex functions of the available invariants, but cannot create new invariants.

## 2 Mathematical Background

We work over a field $k$, typically $\mathbb{R}$ or $\mathbb{C}$. Let $G$ be a linear algebraic group acting rationally on a finite-dimensional vector space $V$. We write $k[V]$ for the ring of polynomial functions on $V$.

### 2.1 Invariants and Relative Invariants

**Definition 2.1** (Invariant). A polynomial $f \in k[V]$ is a $G$-invariant if $f(g \cdot v) = f(v)$ for all $g \in G$ and $v \in V$. The ring of all $G$-invariants is denoted $k[V]^G$.

**Definition 2.2** (Relative Invariant). A polynomial $f \in k[V]$ is a $G$-relative invariant (or semi-invariant) if there exists a character $\chi : G \to k^*$ (a group homomorphism to the nonzero scalars) such that

$$f(g \cdot v) = \chi(g)f(v) \quad \text{for all } g \in G, \, v \in V.$$

An invariant is a relative invariant with trivial character $\chi \equiv 1$. But a relative invariant with non-trivial character is not an invariant.

**Example 2.3.** Let $G = \mathrm{SO}(3) \times \mathbb{R}^*$ act on $V = \mathbb{R}^3$ by $(g, \lambda) \cdot v = \lambda \cdot gv$. The function $f(v) = |v|^2$ satisfies $f((g, \lambda) \cdot v) = \lambda^2 |v|^2 = \chi(g, \lambda)f(v)$ where $\chi(g, \lambda) = \lambda^2$. Thus $f$ is a relative invariant. It is not an invariant: $f(2v) = 4f(v) \neq f(v)$.

For this group, the invariant ring is $\mathbb{R}[V]^G = \mathbb{R}$ (constants only), since any polynomial invariant must satisfy $p(\lambda v) = p(v)$ for all $\lambda \neq 0$, forcing $p$ constant.

### 2.2 Covariants and Equivariant Maps

**Definition 2.4** (Covariant). Let $W$ be another $G$-representation. A polynomial map $\Phi : V \to W$ is a $G$-covariant (or equivariant polynomial) if $\Phi(g \cdot v) = g \cdot \Phi(v)$ for all $g \in G$, $v \in V$.

The set of all covariants $V \to W$, denoted $\mathrm{Cov}_G(V, W)$, forms a module over the invariant ring $k[V]^G$ (meaning covariants can be multiplied by invariant polynomials to get new covariants). When $G$ is reductive (as are all groups considered here), this module is finitely generated: every covariant can be written as a linear combination of finitely many "basic" covariants with coefficients in $k[V]^G$. This is Malgrange's theorem.

### 2.3 Prehomogeneous Vector Spaces

**Definition 2.5.** A representation $(G, V)$ is a prehomogeneous vector space (PVS) if $V$ contains an open $G$-orbit $\Omega$ (a single orbit that is "almost all" of $V$, with complement defined by polynomial equations).

The complement $S = V \setminus \Omega$ is the singular set. For "regular" PVS with reductive $G$, the singular set is a hypersurface defined by a relative invariant.

**Theorem 2.6** (Sato-Kimura Structure Theorem (Sato and Kimura, 1977)). *Let $(G, V)$ be an irreducible regular PVS with $G$ reductive. Then:*

*(i) The singular set $S$ is either empty or a hypersurface defined by a single irreducible polynomial $f$, unique up to scalar.*

*(ii) This f is a relative invariant. All relative invariants with the same character are scalar multiples of powers of f.*

**Remark 2.7** (Invariant Ring for PVS). The invariant ring $k[V]^G$ (polynomials fixed by $G$, not just transforming by a character) depends on the structure of $G$. When $G$ contains scaling transformations $v \mapsto \lambda v$ for $\lambda \in k^*$—as in most PVS relevant to machine learning—the invariant ring is typically $k[V]^G = k$ (constants only). This is because any polynomial satisfying $p(\lambda v) = p(v)$ for all $\lambda \neq 0$ must be constant.

It is tempting to write "$k[V]^G = k[f]$" for a PVS with relative invariant $f$, by analogy with how $\mathbb{R}[|v|^2]$ generates the invariant ring for SO(3) on $\mathbb{R}^3$. This analogy fails: the relative invariant $f$ generates relative invariants (which transform by a character), not invariants. For most PVS relevant to machine learning, the genuine polynomial invariant ring is just constants; all polynomial algebraic content sits in the relative invariants. This is one place where the difference between the compact-group regime (where invariant rings are rich) and the PVS regime (where they are trivial) is easy to miss.

# 3 Analysis of SO(3) on Vectors

We begin with the representation most common in geometric deep learning: SO(3) acting on $\mathbb{R}^3$. This analysis establishes the appropriate tools and corrects a potential misconception.

## 3.1 SO(3) on $\mathbb{R}^3$ is Not a PVS

**Proposition 3.1.** *The representation* $(\mathrm{SO}(3), \mathbb{R}^3)$ *is not a prehomogeneous vector space.*

*Proof.* The orbits of SO(3) on $\mathbb{R}^3$ are spheres $S_r^2 = \{v : |v| = r\}$ for $r > 0$, plus the origin. Each sphere has dimension 2, hence codimension 1 in $\mathbb{R}^3$. There is no orbit of codimension 0, so $(\mathrm{SO}(3), \mathbb{R}^3)$ is not prehomogeneous. $\square$

## 3.2 The First Fundamental Theorem for SO(3)

Since PVS theory does not apply, we use classical invariant theory. The First Fundamental Theorem (FFT) for orthogonal groups completely characterizes the invariant ring and the module of covariants.

**Theorem 3.2** (FFT for SO(3) on $\mathbb{R}^3$). *Let* SO(3) *act on* $V = \mathbb{R}^3$ *by rotations.*

*(i) The invariant ring is* $\mathbb{R}[V]^{\mathrm{SO}(3)} = \mathbb{R}[\rho]$ *where* $\rho = |v|^2$.

*(ii) The module of covariants* $\mathrm{Cov}_{\mathrm{SO}(3)}(\mathbb{R}^3, \mathbb{R}^3)$ *is free of rank 1 over* $\mathbb{R}[\rho]$, *generated by the identity map* $\iota : v \mapsto v$.

This theorem, due to Weyl (1946), has an immediate consequence:

**Corollary 3.3** (Scalar Collapse). *Any* SO(3)*-equivariant polynomial map* $\Phi : \mathbb{R}^3 \to \mathbb{R}^3$ *has the form*

$$\Phi(v) = p(|v|^2) \cdot v \tag{1}$$

*for some polynomial p.*

This is the algebraic origin of scalar collapse: an SO(3)-equivariant network processing a single vector can only modulate the output by functions of the squared norm. The direction of $v$ is preserved (the output is parallel to $v$), but the network cannot discriminate between different directions at the same norm.

**Remark 3.4** (Polynomial vs. Smooth/Continuous Maps). Theorem 3.2 and Corollary 3.3 concern polynomial maps. For $v \neq 0$, the stabilizer $\mathrm{Stab}(v) \cong \mathrm{SO}(2)$ fixes exactly the line $\mathbb{R}v$, so equivariance forces $\Phi(v) \in \mathbb{R}v$ and hence $\Phi(v) = f(v) \cdot v$ for some scalar function $f$. Equivariance implies $f(gv) = f(v)$, so $f$ is SO(3)-invariant.

For continuous invariants on $\mathbb{R}^3$, invariance means $f(v)$ depends only on the orbit parameter $|v|$ (equivalently $|v|^2$), so $\Phi(v) = h(|v|^2) \cdot v$ for a continuous $h$. For smooth invariants, Schwarz's theorem (Schwarz, 1975)

implies $f$ factors smoothly through the polynomial generator $|v|^2$. The polynomial setting captures the essential algebraic structure.

### 3.3 Connection to PVS via Scaling

If we extend $\mathrm{SO}(3)$ to $G = \mathrm{SO}(3) \times \mathbb{R}^*$ acting by $(g, \lambda) \cdot v = \lambda g v$, then $(G, \mathbb{R}^3)$ is a PVS with open orbit $\mathbb{R}^3 \setminus \{0\}$. The function $|v|^2$ becomes a relative invariant (not an invariant), and the invariant ring collapses to constants. This connection explains why PVS ideas are "morally" relevant, but for practical machine learning with pure rotations, the FFT is the correct tool.

## 4 $\mathrm{GL}(n) \times \mathrm{GL}(n)$ on Matrices: A Genuine PVS

Classical FFT-style invariant theory addresses compact-group actions; for non-compact actions like $\mathrm{GL}(n) \times \mathrm{GL}(n)$ on $\mathrm{Mat}_n$, the natural framework is Sato–Kimura prehomogeneous vector space theory (Sato and Kimura, 1977), which classifies representations admitting an open dense orbit and characterizes their relative invariants. This section develops the standard worked example from that classification and traces its consequences for equivariant network expressivity.

### 4.1 Prehomogeneous Structure

We begin by saying what we mean by *rank collapse*. Unlike scalar collapse—where vector information collapses to scalar (length) information—rank collapse refers to the collapse of discriminating power. The key insight is simple: any invertible matrix can be transformed into any other invertible matrix by left and right multiplication. Concretely, given invertible $M$ and $M'$, we have $M' = (M'M^{-1}) \cdot M \cdot I$. This transitivity has a severe consequence: any function that is invariant under the group action must take the same value on all invertible matrices, because they all lie in a single orbit. Therefore, invariant scalar outputs are constant on the entire open orbit—they carry no information about which invertible matrix we have. The situation worsens at lower-rank matrices: powers like $M^2$ cannot be used to build richer features because $(AMB^{-1})^2 \neq AM^2B^{-1}$ in general. The only remaining geometric structure is which rank stratum the matrix lies in.

Consider $G = \mathrm{GL}(n) \times \mathrm{GL}(n)$ acting on $V = \mathrm{Mat}_n(k)$ by $(A, B) \cdot M = AMB^{-1}$.

**Theorem 4.1.** *The representation $(\mathrm{GL}(n) \times \mathrm{GL}(n), \mathrm{Mat}_n)$ is a prehomogeneous vector space with open orbit $\Omega = \mathrm{GL}(n)$ (invertible matrices) and singular set $S = \{M : \det(M) = 0\}$.*

*Proof.* For transitivity on $\mathrm{GL}(n)$: given invertible $M, M'$, we have $M' = (M'M^{-1}, I) \cdot M$. The set $\mathrm{GL}(n)$ is open in $\mathrm{Mat}_n$ (its complement, the singular matrices, is defined by $\det = 0$), so it is the open orbit. □

### 4.2 The Determinant as Relative Invariant

**Theorem 4.2.** *For $(\mathrm{GL}(n) \times \mathrm{GL}(n), \mathrm{Mat}_n)$:*

  (i) *The relative invariant is $f(M) = \det(M)$, with character $\chi(A, B) = \det(A)/\det(B)$.*

  (ii) *The invariant ring is $k[\mathrm{Mat}_n]^G = k$ (constants only).*

*Proof.* For (i): $\det(AMB^{-1}) = \det(A)\det(M)\det(B)^{-1} = \chi(A, B)\det(M)$.

For (ii): Any invariant satisfies $p(\lambda M) = p(M)$ for all $\lambda$ (take $A = \lambda I$, $B = I$), forcing $p$ constant. □

The determinant is a relative invariant, not an invariant. This distinction, emphasized by the Sato-Kimura theory, is essential.

### 4.3 Rank Stratification

The singular set carries a natural stratification by rank.

**Theorem 4.3** (Rank Stratification). *The singular set decomposes as* $S = \bigsqcup_{r=0}^{n-1} \Sigma_r$ *where* $\Sigma_r = \{M : \text{rank}(M) = r\}$. *Each* $\Sigma_r$ *is a single $G$-orbit with* $\text{codim}(\Sigma_r) = (n-r)^2$.

*Proof sketch.* Any rank-$r$ matrix can be transformed via row and column operations (i.e., left and right multiplication by invertible matrices) to the canonical form $\text{diag}(1, \ldots, 1, 0, \ldots, 0)$ with $r$ ones. Hence $G = \text{GL}(n) \times \text{GL}(n)$ acts transitively on $\Sigma_r$. For the codimension: the variety of matrices of rank at most $r$ has dimension $2nr - r^2$ (view rank-$r$ matrices as products $XY^T$ with $X, Y \in \text{Mat}_{n \times r}$, then quotient by the $\text{GL}(r)$ action). Hence $\text{codim}(\Sigma_r) = n^2 - (2nr - r^2) = (n-r)^2$. $\qquad\square$

This stratification, predicted by PVS theory, illuminates rank collapse.

**Lemma 4.4** (Polynomial equivariants for the left–right action are linear). *Let* $G = \text{GL}(n) \times \text{GL}(n)$ *act on* $\text{Mat}_n$ *by* $(A, B) \cdot M = AMB^{-1}$. *If* $\Phi : \text{Mat}_n \to \text{Mat}_n$ *is a polynomial $G$-equivariant map (with the same action on the output), then $\Phi$ is linear.*

*Proof.* Equivariance with $(A, B) = (\lambda I, I)$ implies $\Phi(\lambda M) = \lambda \Phi(M)$ for all $\lambda \in k^*$. Writing $\Phi = \sum_{d \geq 0} \Phi_d$ as a sum of homogeneous polynomial maps of degree $d$, we get $\sum_d \lambda^d \Phi_d(M) = \lambda \sum_d \Phi_d(M)$ for all $\lambda$. Comparing coefficients of $\lambda^d$ forces $\Phi_d = 0$ for $d \neq 1$. Hence $\Phi$ is homogeneous of degree 1, i.e., linear. $\qquad\square$

The key distinctions for rank collapse are:

- Invariant scalars: The invariant ring is $k[\text{Mat}_n]^G = k$ (constants only). An invariant scalar head extracts no information from the input.

- Relative invariants: The determinant $\det(M)$ is a relative invariant defining the singular set. It transforms by a character, so it is not available to invariant computations.

- Covariants: Polynomial $G$-equivariant maps $\Phi : \text{Mat}_n \to \text{Mat}_n$ (with the same left–right action on the output) are extremely limited. The maps $M \mapsto I$ and $M \mapsto M^k$ for $k \geq 2$ are not equivariant: for instance, $(AMB^{-1})^2 = AM(B^{-1}A)MB^{-1} \neq AM^2B^{-1}$ in general. By Lemma 4.4, any polynomial equivariant self-map is linear. Schur's lemma then gives $\Phi(M) = cM$ for some $c \in k$, since $\text{Mat}_n \cong V \otimes V^*$ is irreducible as a $\text{GL}(n) \times \text{GL}(n)$-module over both $\mathbb{R}$ and $\mathbb{C}$.

The severity of rank collapse is now clear: not only does the invariant ring contain no information, but equivariant self-maps are trivial. The only geometric structure is the rank stratification of the singular set.

**Remark 4.5** (Depth for Matrix-Equivariant Networks). This makes depth even less useful for matrix-equivariant networks than for rotation-equivariant networks. For $\text{SO}(3)$ on $\mathbb{R}^3$, the invariant ring $\mathbb{R}[|v|^2]$ is nontrivial, so depth helps approximate complex functions of $|v|^2$. For matrices under $\text{GL}(n) \times \text{GL}(n)$, the invariant ring is just constants, and equivariant layers can only scale: $M \mapsto cM$. Composing such layers yields $M \mapsto c_1 c_2 \cdots c_k M$—still just scaling. There are no nontrivial invariants to build complex functions of, and no nontrivial equivariant transformations to compose.

## 5 Aggregation Escapes the Constraints

Having characterized the limitations of single-fiber processing, we now show how aggregation escapes them.

### 5.1 Product Representations Have Richer Invariants

Let $G$ act on $V$, and consider the product representation $(G, V^n)$ with diagonal action: $g \cdot (v_1, \ldots, v_n) = (g \cdot v_1, \ldots, g \cdot v_n)$.

The invariant ring $k[V^n]^G$ contains not only "fiber invariants" $p(v_i)$ for $p \in k[V]^G$, but also "cross-invariants" depending on multiple $v_i$. When $G$ acts nontrivially, these cross-invariants typically enrich the invariant structure. The following classical result makes this precise for orthogonal groups.

**Theorem 5.1** (O(3) on Multiple Vectors). *For* O(3) *acting diagonally on* $(\mathbb{R}^3)^n$:

$$\mathbb{R}[(\mathbb{R}^3)^n]^{O(3)} = \mathbb{R}\left[\{|v_i|^2\}_{i=1}^n \cup \{v_i \cdot v_j\}_{1 \leq i < j \leq n}\right]. \tag{2}$$

**Remark 5.2** (Generators vs. Algebraic Independence). For $n \leq 3$, the $\frac{n(n+1)}{2}$ generators (norms and dot products) are algebraically independent, so the invariant ring is a polynomial ring. For $n > 3$, these generators satisfy algebraic relations: the Gram matrix $(v_i \cdot v_j)$ has rank at most 3 (since the vectors lie in $\mathbb{R}^3$), so all $4 \times 4$ Gram determinants vanish. The invariant ring is still generated by these dot products, but is not a polynomial ring in $\frac{n(n+1)}{2}$ independent variables.

**Remark 5.3** (SO(3) vs. O(3)). For SO(3) rather than O(3), the invariant ring also includes scalar triple products $[v_i, v_j, v_k] := \det(v_i, v_j, v_k)$ when $n \geq 3$. These are preserved by rotations but change sign under reflections. The main point—that aggregation enriches the invariant structure—holds for both groups.

**Corollary 5.4** (Angles are Accessible). *For two vectors, the angle $\theta$ satisfies $\cos \theta = v_1 \cdot v_2/(|v_1||v_2|)$, which is an orthogonal-group invariant. Networks on pairs of vectors can learn angular relationships.*

**Example 5.5** (Aggregation for Matrices). The same principle applies to the PVS case. For $(GL(n) \times GL(n), \text{Mat}_n^k)$ with $k \geq 2$ matrices under diagonal action $(A, B) \cdot (M_1, \ldots, M_k) = (AM_1B^{-1}, \ldots, AM_kB^{-1})$, new invariants appear. On the open orbit where all matrices are invertible, we have rational invariants (regular functions on $GL(n)^k$, not polynomials on $\text{Mat}_n^k$):

$$\text{tr}(M_i M_j^{-1}) \quad \text{for } i \neq j. \tag{3}$$

To verify: $\text{tr}(AM_iB^{-1} \cdot (AM_jB^{-1})^{-1}) = \text{tr}(AM_iB^{-1} \cdot BM_j^{-1}A^{-1}) = \text{tr}(AM_iM_j^{-1}A^{-1}) = \text{tr}(M_iM_j^{-1})$. These "cross-invariants" are unavailable for single matrices, where even rational invariants on $GL(n)$ are constant. Aggregation escapes PVS constraints just as it escapes FFT constraints.

This explains why message-passing architectures succeed: they aggregate from multiple neighbors, moving from $(G, V)$ to $(G, V^n)$ where richer invariants become accessible.

Theorem 5.1 identifies which invariants suffice to discriminate point clouds, and there is a sharp algorithmic counterpart due to Delle Rose et al. (2023): the geometric $(d-1)$-WL test, initialized with the matrix of mutual distances inside each tuple, is complete for point clouds in $\mathbb{R}^d$ after three iterations, with 2-WL complete in $\mathbb{R}^3$ in particular; the required body-order grows with ambient dimension. Li et al. (2023) make the constructive side of this picture explicit for the distance-matrix setting: their $k$-DisGNN hierarchy attains scalar universality at $k = 2$, matching the FFT prediction that $\mathbb{R}[(\mathbb{R}^3)^n]^{O(3)}$ is generated by norms and pairwise dot products (all recoverable from the Gram matrix), and vector universality at $k = 3$, where parity-odd invariants such as triple products first become available. The body-order split between $k = 2$ for scalars and $k = 3$ for vectors is the algorithmic shadow of the algebraic distinction between $\mathbb{R}[(\mathbb{R}^3)^n]^{O(3)}$ and $\mathbb{R}[(\mathbb{R}^3)^n]^{SO(3)}$. A separate sharp consequence in $d = 3$ is that 1-WL cannot even detect whether a point cloud is planar, since norms together with the unlabeled multiset of pairwise distances do not encode the affine dimension of the point cloud. This is the strongest form of single-fiber scalar collapse—directional information is not merely suppressed, dimensional information about the supporting affine subspace is lost.

The aggregation principle has been realized constructively by Cen et al. (2025), who build a graph-level complete scalar invariant—a "canonical form" $\Gamma(\mathcal{G})$—via four-point positioning on the geometric graph, paired with a full-rank steerable basis set. This yields polynomial-time completeness ($\mathcal{O}(N^2)$ with a learned generator for the reference points) on *asymmetric* geometric graphs. From the perspective developed here, $\Gamma$ is an explicit generating set for the graph-level invariant ring in the regime where the orbit stabilizer is trivial: the four reference points provide enough cross-invariants to coordinatize every configuration up to the diagonal action. The symmetric case they leave open is precisely where the stabilizer is nontrivial; equivariant outputs are then forced into a proper subspace of $\mathbb{R}^{2l+1}$ determined by the stabilizer's representation, obstructing full-rank basis construction at the algebraic level rather than the constructive one. An explicit characterization of when this subspace collapses to zero appears in companion work by Cen et al. (2024),

who reduce the stabilizer obstruction to an explicit trace criterion on the degree-$l$ irrep restricted to $H$. They tabulate the resulting collapse degrees for the standard finite subgroups, giving the explicit algebraic content of the stabilizer obstruction.

Lin et al. (2026) generalize this trace criterion beyond full collapse, classifying the resulting degenerations as full, axial, or half. This is the orbit-type counterpart of asking which cross-invariants in $k[V^n]^G$ remain non-constant on $H$-symmetric configurations. From a purely empirical angle, Li et al. (2023) construct families of symmetric counterexamples that Vanilla DisGNN cannot distinguish despite being non-congruent; these are precisely the high-stabilizer configurations where the trace criterion predicts collapse, providing a concrete experimental manifestation of the algebraic obstruction. Li et al. (2025) sharpen this empirical picture into a structural one: under fully-connected conditions, the set of point clouds that Vanilla DisGNN cannot identify is contained in a subset of Lebesgue measure zero, characterized by coincidence of all sub-cloud geometric centers. They further show that node marking—running DisGNN on each ego-subgraph with one node distinguished—suffices to recover $E(3)$-completeness, because the marked node supplies a second anchor (alongside the geometric center) that breaks the stabilizer and enables triangular distance encoding. Marking is the message-passing realization of the aggregation principle: it converts a single-fiber problem on $V$ into a problem on $V \times \{*\}$ where the additional distinguished coordinate plays the role of a stabilizer-breaking cross-invariant, restoring access to the discriminating ring.

## 5.2 Implications for Modern Equivariant Architectures

The algebraic theory developed above—particularly the transition from $k[V]^G$ to $k[V^n]^G$—directly explains design choices in modern equivariant neural networks. We introduce the concept of body-order to formalize this connection.

**Definition 5.6** (Body-Order)**.** An architecture has body-order $\nu$ at a given layer if its scalar features can be expressed as functions of invariants from $k[V^\nu]^G$—that is, invariants depending jointly on $\nu$ local geometric inputs rather than fiber-wise invariants alone.

For SO(3)-equivariant networks on atomic systems, body-order determines which geometric information is accessible: at $\nu = 1$ only individual distances $|v_i|$ (fiber-wise invariants from $\prod_i k[V]^G$); at $\nu = 2$ distances and angles via $v_i \cdot v_j$ (cross-invariants from $k[V^2]^G$); and at $\nu = 3$ distances, angles, and dihedral-like correlations from $k[V^3]^G$.

We analyze four prominent architectures through this lens:

Table 1: Architecture comparison through invariant ring theory. "Irreps" indicates which SO(3) representations are used ($l = 0$ is scalar, $l = 1$ is vector). "Body-order/layer" is $\nu$ achieved per message-passing layer. "Parity" indicates whether parity-odd (pseudoscalar/pseudovector) features are accessible.

| Architecture | Irreps | Body-order/layer | Parity | Mechanism |
|---|---|---|---|---|
| SchNet | $l = 0$ only | $\nu = 1$ | even only | RBF of distances |
| PaiNN | $l = 0, 1$ | $\nu = 2$ | even only | $\langle \vec{v}, \vec{v} \rangle$ |
| DimeNet | $l = 0$ only | $\nu = 2$ | even only | explicit triplet angles |
| MACE | $l = 0, \ldots, l_{\max}$ | $\nu$ (tunable) | tunable | tensor products |

In Table 1 and the discussion that follows, the body-order $\nu$ is what a *single* message-passing layer accesses—the natural granularity for comparing mechanisms, not a ceiling on what a deep network can represent. Stacking $L$ layers enlarges the receptive field and raises the accessible body-order accordingly (Section 5.3, Remark on Depth vs. Body-Order). SchNet is the extreme case: a single layer reaches only $|v|^2$, and angular information arises only across layers.

SchNet (Schütt et al., 2018) uses only scalar features ($l = 0$) with radial basis functions of pairwise distances. In our framework, SchNet operates in $\prod_i k[|v_i|^2]$—the product of fiber-wise invariant rings. Cross-invariants like $v_i \cdot v_j$ are not directly computed. To access angular information, SchNet must implicitly learn the law

of cosines through network depth, requiring at least 2 layers for a central atom to "see" distances between its neighbors.

PaiNN (Schütt et al., 2021) augments scalars with vector features ($l = 1$). The key operation is the vector inner product in the update block: accumulated vectors $\vec{v}_i = \sum_j w_{ij} \hat{r}_{ij}$ satisfy

$$\langle \vec{v}_i, \vec{v}_i \rangle = \sum_{j,k} w_{ij} w_{ik} (\hat{r}_{ij} \cdot \hat{r}_{ik}) = \sum_{j,k} w_{ij} w_{ik} \cos\theta_{jik}. \tag{4}$$

This directly computes the cross-invariant $v_j \cdot v_k$ from Theorem 5.1, achieving body-order $\nu = 2$ in a single layer. PaiNN's design choice—adding vectors and taking their inner product—is precisely the mechanism that accesses $k[V^2]^G$.

DimeNet (Klicpera et al., 2020) takes an alternative route to body-order 2: explicit enumeration of atom triplets $(k, j, i)$ with direct computation of bond angles $\theta_{kji}$. While DimeNet uses only scalar features, its triplet message-passing accesses $k[V^2]^G$ by construction.

MACE (Batatia et al., 2022) achieves higher body-order through systematic tensor products. The key construction is:
$$A_i^{(\nu)} = A_i^{(1)} \otimes A_i^{(1)} \otimes \cdots \otimes A_i^{(1)} \quad (\nu \text{ factors}) \tag{5}$$
where $A_i^{(1)} = \sum_j R(d_{ij}) Y(\hat{r}_{ij})$ aggregates radial functions times spherical harmonics over neighbors. In our framework, these tensor products create equivariant tensors that encode $\nu$-body correlations; invariant scalar features arise after projecting to $l = 0$ components via Clebsch-Gordan decomposition. The resulting scalars can access $\nu$-body cross-invariants from $k[V^\nu]^G$. The body-order parameter $\nu$ is architecturally tunable, allowing MACE to directly access dihedral-like correlations ($\nu = 3$) or higher without requiring network depth.

PaiNN's vector-inner-product mechanism generalizes to a broader *spherical scalarization* paradigm shared by several recent architectures. The pattern is: build equivariant features $T_i^{(l)} = \sum_j R(d_{ij}) Y^{(l)}(\hat{r}_{ij})$ at angular degree $l$, then form scalars by inner products $\langle T_i^{(l)}, T_j^{(l)} \rangle$. Cen et al. (2024)'s HEGNN makes this paradigm explicit and argues that high-degree representations are necessary, not optional: their trace criterion identifies which degrees collapse on $H$-symmetric inputs, and distinguishing such configurations requires channels at degrees that escape it. From the invariant-ring perspective, the spherical-harmonic addition theorem gives $\langle Y^{(l)}(\hat{r}), Y^{(l)}(\hat{r}') \rangle \propto P_l(\hat{r} \cdot \hat{r}')$, so $\langle T_i^{(l)}, T_j^{(l)} \rangle$ accesses cross-invariants $\hat{r} \cdot \hat{r}'$ from $k[V^2]^{O(3)}$ filtered through the degree-$l$ Legendre polynomial—higher $l$ resolves finer angular structure but stays within the O(3)-invariant subring. Aykent and Xia (2025)'s GotenNet refines this further by combining geometry-aware tensor attention with inner-product-based edge refinement across degrees, achieving state-of-the-art molecular benchmark performance while remaining within $k[V^n]^{O(3)}$. PaiNN ($l = 1$ only), HEGNN (all $l$), and GotenNet (attention over $l$) thus form a refinement hierarchy at fixed body-order $\nu = 2$ and fixed parity (even): each strengthens approximation power within the parity-even subring, but none crosses into the parity-odd subring required for chirality.

The body-order claims above admit a complementary algorithmic justification. Li et al. (2023) prove that DimeNet is expressible as a 2-DisGNN—a $k$-WL or $k$-FWL message-passing scheme acting on the all-pair distance matrix at $k = 2$—and that GemNet, which incorporates dihedral terms, is expressible as a 3-DisGNN. In our language, these reductions confirm constructively that DimeNet's triplet construction accesses $k[V^2]^G$ and that dihedral-aware architectures access $k[V^3]^G$, giving an explicit path from invariant-ring access to message-passing implementation. The same hierarchy explains why their $k$-DisGNN framework achieves scalar universality only once $k \geq 2$: $k = 1$ (Vanilla DisGNN) sees only the fiber-wise invariants $\prod_i k[|v_i|^2]$, which by Section 3 cannot discriminate any SO(3) orbit beyond the radial parameter.

Li et al. (2025) provide a benchmark of synthetic counterexamples with that symmetry property that Vanilla DisGNN cannot distinguish by construction. They report that PaiNN and MACE distinguish all pairs, alongside the four invariant architectures (DimeNet, SphereNet, GemNet, GeoNGNN) whose completeness they prove formally; they do not provide a theoretical account of PaiNN's and MACE's success. Body-order analysis offers one such account: PaiNN reaches $\nu = 2$ in a single layer via $\langle \vec{v}_i, \vec{v}_i \rangle$, accessing the cross-invariants $\hat{r}_{ij} \cdot \hat{r}_{ik}$ that resolve high-stabilizer configurations, and MACE's tensor products reach $\nu \geq 2$ by

construction. The mechanism behind their empirical separation success is the same body-order $\geq 2$ access to cross-invariants that makes DimeNet and GemNet succeed, even if implemented through different machinery.

*Parity and chirality.* The invariant rings $k[V^n]^{O(3)}$ and $k[V^n]^{SO(3)}$ differ: the latter includes pseudoscalars like $\det(v_1, v_2, v_3)$. Architectures restricted to parity-even scalar features—effectively O(3)-invariant readouts—provably cannot represent pseudoscalars and therefore cannot detect chirality. Architectures that include parity-odd (pseudo) channels, such as MACE configured with parity tracking through Clebsch-Gordan decompositions, can access the full SO(3) invariant ring and detect chirality. This is a direct consequence of which invariant ring the architecture can access, not a limitation of specific named models per se (implementations vary in whether they include parity-odd features). The same dichotomy appears constructively in Cen et al. (2025), whose chirality experiments contrast a determinant-based pseudoscalar with a tensor-product construction. At the tensor-product-operation level, Xie et al. (2025) identify the same obstruction via selection rules: the Gaunt tensor product (GTP) requires $\ell_1 + \ell_2 + \ell_3$ even and so excludes the $1o \otimes 1o \to 1e$ path (the cross product); they confirm experimentally that GTP-based networks cannot distinguish chiral 3D structures regardless of depth or feature degree. Xie et al. (2026) subsequently show that this obstruction is fixable at the operation level: their vector signal tensor product (VSTP) retains the asymptotic speedup of GTP while admitting the parity-odd paths. What determines chirality detection is access to the parity-odd subring of $k[V^n]^{SO(3)}$, not the speed of the underlying tensor product operation. The same correspondence is visible at the message-passing level in Li et al. (2025): their fully-connected DisGNN-based models (DimeNet, SphereNet, GemNet, GeoNGNN) attain $E(3)$-completeness but cannot detect chirality, since pairwise Euclidean distance is O(3)-invariant. To distinguish enantiomers, they introduce GeoNGNN-C, which augments the inner-subgraph edge feature with the sign of the triple product $\vec{r}_{ci} \times \vec{r}_{cj} \cdot \vec{r}_{ck}$, where $c$ is the geometric center and $k$ is the subgraph center. The injected sign is the parity-odd pseudoscalar from Theorem 5.1—the generator of $\mathbb{R}[(\mathbb{R}^3)^n]^{SO(3)} \setminus \mathbb{R}[(\mathbb{R}^3)^n]^{O(3)}$—and their SE(3)-completeness result is then the algebraic prediction: parity-even distance-based aggregation cannot reach the parity-odd subring, and supplying one pseudoscalar at the input restores it.

## 5.3 Boundaries: Depth, Sparsity, and Input Symmetry

**Remark 5.7** (Depth vs. Body-Order)**.** Our analysis clarifies when depth helps and when it cannot. The typical scaling is as follows: for SchNet ($\nu = 1$ per layer), $L$ layers can access up to $L$-body correlations, since each layer propagates distance information one hop further. For PaiNN/DimeNet ($\nu = 2$ per layer), depth can provide up to $(L+1)$-body correlations. For MACE with tunable $\nu$, depth can provide up to $L \cdot \nu$-body correlations. These are upper bounds; the precise body-order achieved depends on the specific operations and how information combines across layers. However, if a target function requires invariants outside the architecture's accessible ring (e.g., chirality for parity-even networks), no amount of depth suffices. This is consistent with Theorem 6.1, which fixes the input fiber: depth never enlarges $k[V]^G$. Stacking layers enlarges a different object, the set of fibers being aggregated, because each additional hop reaches a wider neighborhood. The effective body-order, the largest $n$ for which the readout accesses $k[V^n]^G$, then grows with the number of hops. What lifts body-order is aggregation, not depth applied to a fixed fiber. This is consistent with Theorem 6.1, which fixes the input fiber: depth never enlarges $k[V]^G$. Stacking layers does something different—it enlarges the *set* of fibers being aggregated, since each hop draws in further neighbors—so the effective body-order, the largest $n$ for which the readout reaches $k[V^n]^G$, grows with the number of hops. The mechanism is aggregation, not depth on a fixed fiber.

The body-order analysis above fixes an architecture and asks which input distinctions it can resolve. A complementary practical question is the reverse: given an input with known stabilizer $H$, which feature degrees should the architecture include? This is addressed by Lin et al. (2026), building on the partial-degeneration spectrum from Section 5. For every closed subgroup $H \subseteq O(3)$ and every irrep degree $l_0$, they tabulate whether $(H)$ is realizable as an orbit type in $V_{l=l_0^\pm}^{\oplus r}$, and when it is not, the resulting symmetry infimum $I_G(V_{l=l_0^\pm}^{\oplus r}, H)$. Channels whose infimum is the full group on inputs of stabilizer type $H$ suffer full degeneration: their invariant output is constant on the $G$-orbit and contributes only constants to the readout. Their experiments confirm the practical consequence—including such fully-degenerate channels can be actively detrimental for molecules with the corresponding symmetry, consistent with the algebraic

prediction that those channels contribute no discriminating cross-invariants from $k[V^n]^G$ for $H$-symmetric inputs.

A separate caveat applies when the message-passing graph is sparse rather than full. The body-order analysis implicitly assumes each node can aggregate over every other node, and the topology of aggregation interacts with body-order in a way the invariant-ring algebra alone does not capture. Sverdlov and Dym (2025) characterize this interaction sharply: on a sparse combinatorial graph $A$, an architecture with scalar-only intermediate features (which they call I-GGNN, and which includes SchNet on the input graph) is generically complete if and only if $A$ is generically globally rigid in $\mathbb{R}^d$, while an architecture with equivariant vector features (E-GGNN, including EGNN, TFN, GVP, and MACE) is generically complete if and only if $A$ is connected, with depth $T \geq d + 1$ sufficient. From the invariant-ring perspective the dichotomy is sharp. An I-GGNN on $A$, regardless of depth, has scalar features that are continuous functions of the edge squared distances $\{|v_i - v_j|^2 : (i, j) \in E(A)\}$. Composition across hops distributes these scalars across the graph but cannot enlarge the generating set, by the same depth-independence argument as Theorem 6.1(c). Whether the resulting subring separates generic orbits is therefore a purely combinatorial question—do generic edge lengths determine the configuration up to congruence?—which is exactly the definition of generic global rigidity of $A$ in $\mathbb{R}^d$. An E-GGNN, by contrast, propagates equivariant vector features that carry relative orientation across each edge; $d + 1$ layers on a connected $A$ suffice to assemble a global frame, after which cross-invariants not supported on $E(A)$ become reconstructible. SchNet thus inherits a topological constraint on the underlying graph that the equivariant-feature architectures do not. The constructive fix—applying I-GGNN to $G^{d+1}$, known to be generically globally rigid for any connected $G$—is the topological counterpart of our aggregation principle: scalar-only architectures pay for the absence of equivariant intermediate features by inflating the effective aggregation radius, accessing higher-body-order cross-invariants from $k[V^\nu]^G$ through additional hops rather than through richer per-layer features.

The depth, parity, symmetry-infimum, and sparsity considerations above all point to a single practical principle: architecture choice should match task requirements. Angle-dependent properties require $\nu \geq 2$ (ruling out shallow SchNet), dihedral-dependent properties benefit from $\nu \geq 3$ (favoring MACE over PaiNN/DimeNet at equivalent depth), and chiral properties require parity-odd features (ruling out any parity-even-only configuration).

## 6 The Role of Depth

A natural question is whether depth can compensate for single-fiber limitations. It cannot.

**Theorem 6.1** (Depth Does Not Add Invariants). *Consider a $G$-equivariant neural network with equivariant linear layers, pointwise nonlinearities applied only to scalar (invariant) channels, tensor products, and an input layer that extracts generating invariants of $k[V]^G$. For invariant scalar outputs:*

   *(a) Points on the same $G$-orbit cannot be separated, regardless of depth.*

   *(b) The network output is a function of the generating invariants of $k[V]^G$. For polynomial activations, this is a polynomial in the generators; for continuous activations, a continuous function of them.*

   *(c) Depth allows approximating more complex functions of these generating invariants, but does not create new algebraically independent invariant features.*

*Proof.* (a) If $v' = g \cdot v$ for some $g \in G$, then for any $G$-invariant function $\phi$ we have $\phi(v') = \phi(g \cdot v) = \phi(v)$ by definition. Since the network computes a $G$-invariant output (by equivariance of intermediate layers and invariance of the final scalar), it cannot distinguish $v$ from $v'$.

(b) We proceed by induction on layers. By assumption, the input layer extracts the generating invariants of $k[V]^G$ as scalar features. At each subsequent layer: (i) equivariant linear maps preserve the property that scalar channels contain invariants; (ii) tensor products followed by Clebsch-Gordan (CG) decomposition (the standard method to decompose tensor products into irreducible representations) produce new channels, but the scalar components remain invariants; (iii) pointwise nonlinearities $\sigma$ applied to scalar channels produce

$\sigma(p_1, \ldots, p_k)$ where $p_i$ are functions of the generating invariants—this remains a function of those generators. By induction, the final output is a function (polynomial or continuous, depending on $\sigma$) of the generating invariants.

(c) For polynomial networks: the invariant ring $k[V]^G$ has a fixed transcendence degree (the number of algebraically independent generators) over $k$, determined by representation theory. Polynomial composition cannot increase transcendence degree, so depth cannot create new algebraically independent invariants. For continuous networks: MLPs with continuous activations can approximate arbitrary continuous functions of their inputs (universal approximation), but the inputs to the MLP are still only the fixed set of generating invariants. Depth improves approximation quality within the class of continuous functions of these generators. $\square$

The hypotheses describe an idealized architecture class. Real networks may extract only a subset of $k[V]^G$'s generators in the input layer (SchNet, for instance, extracts $|v|^2$ via radial basis functions but not higher polynomial generators separately); in that case the theorem's conclusion applies to the extracted subset, and depth cannot recover the omitted generators. Theorem 6.1 concerns a *fixed* input fiber $V$; it does not constrain message-passing networks, whose depth enlarges the aggregated neighborhood and with it the effective body-order. That distinct mechanism is treated in Section 5.3.

# 7 Experimental Validation

Two consequences of the algebraic account admit direct experimental illustration: (1) networks with different symmetry groups but the same architecture template exhibit qualitatively different expressivity, and (2) aggregation enables learning that per-fiber processing cannot achieve. These are not predictive tests of the framework on real benchmarks—they verify that the algebraic obstructions remain in force under finite-sample optimization. Code is provided as supplementary.

## 7.1 Setup

All networks are MLPs with two hidden layers of 64 units and ReLU activations, operating on hand-computed invariant features. Training uses Adam with learning rate $10^{-3}$ for 1000 epochs on 10,000 samples, with evaluation on 2,000 held-out samples. We report $R^2$ scores; $R^2 = 1$ indicates perfect prediction, $R^2 = 0$ indicates prediction of the mean, and $R^2 < 0$ indicates worse-than-mean predictions.

We sample vectors from an isotropic distribution: directions are uniform on $S^2$, and radii are drawn independently from a distribution bounded away from zero. For Experiment 1, we additionally ensure reflection symmetry by including both each triple $(v_1, v_2, v_3)$ and its reflection, so parity-even features are unchanged while the triple product flips sign. These distributional choices make the "impossibility" claims that follow mathematically precise rather than heuristic.

## 7.2 Experiment 1: $\mathrm{SO}(3)$ vs $\mathrm{O}(3)$ on the Triple Product

Motivation. The triple product $\det(v_1, v_2, v_3) = v_1 \cdot (v_2 \times v_3)$ is a pseudoscalar: it is $\mathrm{SO}(3)$-invariant but changes sign under reflection, so it is not $\mathrm{O}(3)$-invariant. The invariant rings have different generators: $\mathbb{R}[(\mathbb{R}^3)^3]^{\mathrm{O}(3)}$ is generated by norms $|v_i|^2$ and dot products $v_i \cdot v_j$, while $\mathbb{R}[(\mathbb{R}^3)^3]^{\mathrm{SO}(3)}$ is generated by norms, dot products, and the triple product (subject to algebraic relations). This provides a clean test: both networks process three vectors with identical MLP architectures, differing only in which invariants they compute.

Architectures. The $\mathrm{SO}(3)$-invariant network computes 7 features: three squared norms $|v_i|^2$, three dot products $v_i \cdot v_j$, and the triple product $\det(v_1, v_2, v_3)$. The $\mathrm{O}(3)$-invariant network computes only the 6 parity-even features, excluding the triple product.

Target. $f(v_1, v_2, v_3) = \det(v_1, v_2, v_3)$.

Results.

| Network | Test MSE | $R^2$ |
|---|---|---|
| SO(3)-invariant (7 features) | $4 \times 10^{-6}$ | 1.000 |
| O(3)-invariant (6 features) | 1.76 | $-0.308$ |

**Interpretation.** The SO(3) network learns the target perfectly because the triple product is in its feature set. The O(3) network achieves $R^2 < 0$, performing worse than predicting zero. Under our reflection-symmetric sampling, any function of parity-even invariants has zero correlation with the parity-odd target: $\mathbb{E}[\det(v_1, v_2, v_3) \mid \text{parity-even features}] = 0$. Thus the Bayes-optimal predictor given only parity-even features is the constant 0, and the fundamental limit is $R^2 = 0$. The negative $R^2$ is a finite-sample optimization artifact where the network fits spurious correlations that fail to generalize. This is not a failure of architecture or optimization but a mathematical impossibility: no function of parity-even invariants can correlate with a parity-odd target under reflection-symmetric data. The same impossibility appears in Xie et al. (2025), who show that a Gaunt-tensor-product message-passing network plateaus at chance accuracy when asked to classify chiral 3D Tetris pieces, while a Clebsch-Gordan variant solves the task perfectly—a different empirical setup (classification with learned features, varying depth and $L_{\max}$) but the same algebraic obstruction.

### 7.3 Experiment 2: Aggregation Escapes Scalar Collapse

**Motivation.** Theorem 5.1 predicts that the invariant ring $\mathbb{R}[(\mathbb{R}^3)^2]^{O(3)}$ contains cross-invariants $v_1 \cdot v_2$ that are unavailable to per-fiber processing. (For $n = 2$, the SO(3) and O(3) invariants coincide since there is no triple product.) We test whether this algebraic difference translates to a learning difference.

**Architectures.** The norm-only network processes each vector independently: it computes $|v_1|^2$ and $|v_2|^2$, passes each through a shared MLP, then combines the outputs. This simulates a network without message-passing. The cross-invariant network computes all three invariants $|v_1|^2$, $|v_2|^2$, $v_1 \cdot v_2$ jointly.

**Target.** $f(v_1, v_2) = \cos\theta = (v_1 \cdot v_2)/(|v_1||v_2|)$, defined on $\{v_1 \neq 0, v_2 \neq 0\}$; we sample vectors with norms bounded away from zero. This is a continuous invariant (involving division), not a polynomial, but continuous invariants on $\mathbb{R}^3$ depend only on the orbit parameters and hence are functions of the polynomial generators.

**Results.**

| Network | Test MSE | $R^2$ |
|---|---|---|
| Norm-only (no cross-invariants) | 0.342 | $-0.023$ |
| Cross-invariant (has $v_1 \cdot v_2$) | $2.1 \times 10^{-5}$ | 0.9999 |

**Interpretation.** The norm-only network cannot learn angles because angular information is discarded before combination—this is scalar collapse at the per-fiber level. Under isotropic sampling, $\mathbb{E}[\cos\theta \mid |v_1|, |v_2|] = 0$, so no function of $(|v_1|^2, |v_2|^2)$ can predict $\cos\theta$ better than a constant baseline. The cross-invariant network succeeds because $v_1 \cdot v_2$ is directly available. The gap is not gradual: it is perfect learning versus complete failure, matching the algebraic prediction that angles are functions of cross-invariants but not of fiber-wise norms alone.

Both experiments confirm sharp, qualitative predictions of the theory: different symmetry groups yield different invariant rings, causing identical architectures to exhibit categorically different expressivity (SO(3) vs O(3)), and aggregation enriches the invariant ring, enabling learning that per-fiber processing provably cannot achieve. These results go beyond "invariant networks cannot learn non-invariant targets" (which is true by construction) to demonstrate that the specific structure of invariant rings—which invariants are present, not just that some exist—determines what can be learned.

# 8 Discussion

## 8.1 When Different Tools Apply

Our analysis clarifies when different mathematical frameworks are appropriate. Classical (compact-group) invariant theory applies to actions like SO(3) on $\mathbb{R}^3$: the FFT generates a non-trivial invariant ring ($\mathbb{R}[|v|^2]$) and expressivity follows from covariant module structure. Sato–Kimura PVS theory applies to non-compact actions admitting an open dense orbit, where classical FFTs do not extend; $\mathrm{GL}(n) \times \mathrm{GL}(n)$ on matrices is the standard worked example, with the polynomial invariant ring collapsing to constants and the algebraic content sitting in the relative invariant $\det(M)$ and its rank stratification. General invariant theory governs aggregation in both regimes: moving from $V$ to $V^n$ enriches invariants regardless of which framework governs the single fiber.

## 8.2 Connection to Universality and Separation Results

Our algebraic framework complements and clarifies several lines of work on neural network expressivity.

Universal approximation. Keriven and Peyré (2019) and Yarotsky (2022) establish that sufficiently expressive equivariant architectures can approximate any continuous equivariant function. Our results are consistent: Theorem 6.1 shows that networks can approximate arbitrary functions of the generating invariants. For the specific case of SO(3) on $\mathbb{R}^3$, the orbit space is one-dimensional (parameterized by $|v|$), so continuous invariants are automatically functions of $|v|^2$; for smooth invariants of compact Lie groups more generally, Schwarz's theorem (Schwarz, 1975) guarantees they factor through polynomial generators. The limitation is not approximation power but the input features: if the invariant ring is $k$ (constants), there is nothing to approximate.

Separation and discrimination. Maron et al. (2019a) and Xu et al. (2019) characterize when graph neural networks can distinguish non-isomorphic graphs, connecting to the Weisfeiler-Leman hierarchy. For geometric graphs, Joshi et al. (2023) extend this through the GWL framework. As discussed mechanistically in Section 5, the sharp completeness thresholds of Delle Rose et al. (2023) and Sverdlov and Dym (2025), the body-order universality of Li et al. (2023), the near-completeness and node-marking results of Li et al. (2025), and the canonical-form construction of Cen et al. (2025) all operationalize a single algebraic constraint that Theorem 6.1(a) makes precise: points on the same $G$-orbit cannot be separated by invariant outputs, regardless of architecture details. This also explains why the symmetric-graph case left open by Cen et al. (2025) is structurally harder: nontrivial orbit stabilizers force equivariant maps into a proper subspace of $\mathbb{R}^{2l+1}$, characterized algebraically by the trace criterion of Cen et al. (2024) and extended to the full symmetry-increase spectrum by Lin et al. (2026).

Higher-order methods. The success of higher-order graph networks (Maron et al., 2019b; Morris et al., 2019) and $k$-WL methods parallels our aggregation results: moving from single-node to $k$-tuple processing enriches the invariant structure, just as $(G, V) \to (G, V^n)$ enriches $k[V]^G$ to $k[V^n]^G$ with cross-invariants.

Tensor product operations. The same correspondence appears at the level of individual bilinear operations within an equivariant network. Xie et al. (2025) compare the Clebsch-Gordan, Gaunt, and matrix tensor products through selection rules; their analysis is the operation-level shadow of our invariant-ring picture, with each tensor product opening a particular set of paths into $k[V^n]^G$ and faster variants generally closing some of them. The vector signal tensor product of Xie et al. (2026) restores essentially all the paths while retaining the asymptotic speedup. The scalarization architectures GotenNet (Aykent and Xia, 2025) and its HEGNN predecessor (Cen et al., 2024), discussed in Section 5.2, sit on the same map: they trade off mechanism cost against which cross-invariants in $k[V^n]^{\mathrm{O}(3)}$ they can reach.

Geometric deep learning. Bronstein et al. (2021) provide a programmatic framework for constructing equivariant architectures around symmetry principles—the "GDL blueprint" of interleaving equivariant linear maps with equivariant pointwise nonlinearities. Our contribution is complementary: rather than prescribing how to build equivariant networks, we characterize which functions they can and cannot represent, via the algebraic structure of the relevant invariant rings. Classical FFTs and Sato–Kimura PVS theory provide the algebraic tools to make these constraints precise.

### 8.3 Limitations

Our analysis concerns polynomial maps; extending to smooth or continuous maps requires additional tools such as Schwarz's theorem. Finite groups like $S_n$ are better handled by the Weisfeiler-Leman framework than by classical invariant theory. We have characterized which functions are representable but not how gradient descent navigates the space of equivariant functions—learning dynamics remain to be studied. The experiments in Section 7 are illustrative rather than predictive: they verify that algebraically forbidden discriminations remain forbidden under finite-sample optimization, but do not test the framework on real benchmarks. Validating body-order-driven architectural choices on chemical, materials, or point-cloud datasets—and identifying tasks where the framework predicts a non-obvious architectural preference—is the natural next step.

## 9 Conclusion

We asked why certain group representations lead to expressivity limitations in equivariant neural networks, and found that the answer lies in the algebraic structure of invariant rings.

The case of SO(3) acting on $\mathbb{R}^3$ illustrates the mechanism in the compact-group regime. The First Fundamental Theorem tells us that the invariant ring is generated by $|v|^2$ alone, and consequently every equivariant map $\mathbb{R}^3 \to \mathbb{R}^3$ has the form $v \mapsto p(|v|^2) \cdot v$. Directional information is not suppressed by accident or poor architecture; it is algebraically inaccessible. The case of $\mathrm{GL}(n) \times \mathrm{GL}(n)$ on matrices is more severe: the action is non-compact and lies outside the reach of compact-group FFTs, and Sato–Kimura PVS theory shows the polynomial invariant ring contains only constants, with equivariant self-maps reducing to scalar multiples of the identity. The rank stratification of the singular set is all the geometric structure that remains.

The escape from these constraints comes not from depth but from aggregation. When we pass from a single fiber $V$ to the product $V^n$, the invariant ring acquires cross-invariants that encode relationships between fibers. For orthogonal groups, these are the dot products $v_i \cdot v_j$ that give access to angles; for $n \geq 3$, the triple products $\det(v_i, v_j, v_k)$ give access to chirality. This algebraic fact explains why message-passing architectures succeed where per-edge processing fails, and why body-order—the number of geometric inputs combined before extracting invariants—is the relevant architectural parameter for expressivity.

For practitioners, the implication is concrete: when performance plateaus, check the invariant ring before adding layers. Depth improves approximation of functions that are already representable; it cannot conjure invariants that the algebra forbids.

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
