# OpenReview forum: "Why Equivariant Networks Lose Information: Invariant Rings and the Role of Aggregation"
_TMLR — Rejected by TMLR_

### Review · Reviewer_911M · 2026-03-07

**Summary Of Contributions:**

The paper’s main contribution is an algebraic explanation of **why some equivariant networks are fundamentally limited in what they can represent**. In particular, it argues that for certain group actions, the available invariants are too poor for depth alone to recover missing information. The paper’s central message is that **aggregation over multiple objects creates new cross-invariants**, while stacking more layers on a single object does not. It then uses this viewpoint to interpret common geometric architectures and gives small experiments that match the theory.

**Strengths:** The paper has a clear conceptual message, a strong mathematical framing, and a useful interpretation of why higher-order or multi-body interactions help in practice. It also connects the theory to well-known architectures rather than staying purely abstract.

**Weaknesses:** The empirical section is fairly limited and mostly toy-scale, so the practical impact is argued more than demonstrated. Also, some of the high-level insights may feel familiar to researchers in equivariant ML, making the novelty depend heavily on the specific algebraic framing.

**Audience:**

Yes

**Audience Explanation:**

Yes. I think at least a meaningful subset of TMLR’s audience would be interested in this paper, especially researchers working on equivariant learning, geometric deep learning, molecular ML, and expressivity theory. The paper addresses a real and fairly central question—why certain equivariant architectures lose information, and why aggregation can recover expressivity that depth alone cannot—and it connects that question directly to architectures such as SchNet, PaiNN, DimeNet, and MACE.

6979_Why_Equivariant_Networks_



6979_Why_Equivariant_Networks_

It also seems well aligned with TMLR’s readership because it sits at the intersection of theory and modern ML practice: it frames the problem using invariant theory and PVS theory, while explicitly positioning itself relative to universality, Weisfeiler–Leman-style expressivity analyses, and practical geometric architectures. That combination should be of interest to readers who care about principled architectural design, even if the paper is probably more relevant to a specialized segment of TMLR than to the entire community.

6979_Why_Equivariant_Networks_



6979_Why_Equivariant_Networks_

**Broader Impact Concerns:**

I do not have significant ethical concerns about this work. It is primarily a theoretical/mathematical paper about expressivity in equivariant networks, and I did not see any direct dual-use, privacy, or fairness risks that would clearly require a substantial Broader Impact discussion. At most, the authors could briefly note that the work may influence model design in application areas such as science or chemistry, but I do not view this as essential.

**Claims And Evidence:**

Yes

**Claims Explanation:**

**Partly yes.** The theoretical claims are generally supported by clear algebraic arguments, and the paper does a good job distinguishing what is formally proved from the intuition. In particular, the claims about single-fiber expressivity limits, the difference between the SO(3) and matrix-equivariant settings, and the role of aggregation are backed by explicit invariant-theoretic reasoning and concrete propositions/lemarks.

That said, the **empirical evidence is much narrower** than the breadth of the paper’s overall message. The experiments appear mainly as controlled toy demonstrations confirming the qualitative predictions of the theory, rather than strong evidence that the proposed perspective yields clear advantages on realistic architectures or benchmark tasks. So the paper is convincing as a **theory/explanation paper**, but less convincing if one expects broad empirical validation of the practical claims.

I would therefore say: **the theoretical claims are mostly well supported and clearly argued, while the practical/architectural implications are suggestive rather than fully demonstrated.**

**Requested Changes:**

Please cite and discuss the following closely related work:

Sverdlov, Y. and Dym, N. On the Expressive Power of Sparse Geometric MPNNs. arXiv:2407.02025, 2024.

A short comparison to this paper would help clarify how the present submission relates to prior results on geometric message-passing expressivity, especially regarding sparse connectivity, equivariant versus invariant intermediate features, and separation power.

---

> ### Author Response · Authors · 2026-05-20
>
> Thank you for the constructive read. Both points are addressed in the revision.
>
> **Sverdlov & Dym (2024).** Now in §1.3 and a dedicated paragraph in §5.3 (Boundaries: Sparsity). The invariant-ring reading is direct: an I-GGNN on a sparse $A$ has scalar features in the subring generated by edge squared distances $\{|v_i-v_j|^2 : (i,j)\in E(A)\}$, and Theorem 6.1(c) says composition across hops cannot enlarge this generating set. Whether the I-GGNN separates generic orbits then reduces to whether $A$ is generically globally rigid in $\mathbb{R}^d$ — exactly their characterization. An E-GGNN carries relative orientation in equivariant intermediate features, so $d+1$ layers on a connected $A$ suffice to assemble a global frame. Their fix of running I-GGNN on $G^{d+1}$ is the topological counterpart of our aggregation principle: scalar-only architectures pay for the missing equivariant features by inflating aggregation radius. The result fits cleanly and improves the paper.
>
> **Empirical scope.** Agreed. The Limitations section now states explicitly that the experiments are illustrative rather than predictive — controlled verifications that the algebraic obstructions remain binding under finite-sample optimization, not benchmark validation. Impossibility claims about which invariants a symmetry group admits are established by algebra; the experiments verify the obstruction remains in force under finite-sample optimization.
>
> Two pieces of larger-scale external validation in the revision probe the framework's predictions on real architectures:
>
> 1. **Chirality.** Xie, Daigavane, Kotak, Smidt (ICML'25): Gaunt-tensor-product networks plateau at chance on chiral 3D Tetris regardless of depth or $L_{\max}$; Clebsch-Gordan variants succeed. Same parity-odd obstruction as Experiment 1, in their setting with learned features.
>
> 2. **Body-order.** Li, Wang, Kang, Zhang (ICLR'25): on symmetric counterexamples, PaiNN and MACE distinguish all pairs; Vanilla DisGNN fails. Matches Table 1.
>
> A body-order study on chemical/materials benchmarks is the natural follow-up and is flagged in Limitations as ongoing work.

---

> > ### Comment · Reviewer_911M · 2026-05-20
> > **Final decidion**
> >
> > After reading, I feel there's no need for more improvement.

---

### Review · Reviewer_dJpL · 2026-04-17

**Summary Of Contributions:**

Honestly I'm a bit at loss at what to write here. Most of the paper is so far removed from the usual ML and geometric deep learning language that I have trouble identifying notions that I know, to properly express the contributions of this paper.

From what I get, the authors distinguish when PVS theory is applicable or not to some instances of group invariance, or when specialized theorems like the FFT must be used. They claim to study the role of "aggregation" and that it allows to "escape" some kind of degeneracy, but I'm actually incapable to tell if the objects that they are looking have anything to do with actual architectures in geometric deep learning.

**Audience:**

No

**Audience Explanation:**

Obviously the paper appeals to a very niche community, but even then the links with actual machine learning and geometric deep learning are undecipherable for all but a fraction of TMLR's audience.

**Claims And Evidence:**

No

**Claims Explanation:**

I have no idea. The paper does not contain a single actual geometric deep learning architecture described simply.

**Requested Changes:**

In my opinion, the paper should be reworked entirely to:
- clarify the links between the developed theory and actual deep learning architectures, with notations and notions that the community knows
- clarify the actual contribution: for now I have no idea what the authors actually prove, if this is novel for the examples examined in the paper, or if this has any implication in practice (or even in theory, when expressed in layman's terms)

The approach should be pedagogical and slower-paced: for now the paper is extremely dense, with seemingly unrelated results all over the place with no explanation to their implications, or their links with actual ML.

---

> ### Author Response · Authors · 2026-05-20
>
> We thank the reviewer and take seriously the concern that the bridge to standard geometric deep learning may not be visible enough.
>
> **Concrete architectural analysis is already in the paper.** Section 5.2 and Table 1 compare four standard equivariant message-passing architectures — SchNet, PaiNN, DimeNet, MACE — along three axes the framework predicts matter: which $\mathrm{SO}(3)$ irreps the network uses, which body-order each layer reaches, and whether parity-odd features are accessible. The revision extends the §5.2 analysis to HEGNN and GotenNet. The PaiNN analysis is built around the actual PaiNN update step,
>
> $$\langle \vec v_i, \vec v_i\rangle \;=\; \sum_{j,k} w_{ij}w_{ik}(\hat r_{ij}\cdot \hat r_{ik}) \;=\; \sum_{j,k} w_{ij}w_{ik}\cos\theta_{jik},$$
>
> identified as the mechanism that accesses the cross-invariant $v_j\cdot v_k\in\mathbb{R}[V^2]^{\mathrm{O}(3)}$ in a single layer. The MACE analysis is built around the actual MACE tensor product $A^{(\nu)}_i = A^{(1)}_i\otimes\cdots\otimes A^{(1)}_i$ ($\nu$ factors), identified as the mechanism reaching body-order $\nu$ without recourse to depth. SchNet's features live in $\prod_i \mathbb{R}[|v_i|^2]$, which makes precise the well-known fact that SchNet needs depth to learn the law of cosines. These are equations from the original papers with our framework attached to explain why they have the expressivity properties they have.
>
> **The contribution.** The theorems we use are classical (Weyl 1946; Sato–Kimura 1977) and we are explicit about this. What is new is their application:
>
> 1. **Architecture-specific expressivity has an algebraic source.** PaiNN reaches body-order 2 in one layer because $\hat r_{ij}\cdot\hat r_{ik}$ generates $\mathbb{R}[V^2]^{\mathrm{O}(3)}$; SchNet cannot, because its features live in the product of fiber-wise rings; no parity-even architecture detects chirality, because $\det(v_1,v_2,v_3)$ generates exactly the part of $\mathbb{R}[V^3]^{\mathrm{SO}(3)}$ that is *not* in $\mathbb{R}[V^3]^{\mathrm{O}(3)}$.
>
> 2. **Body-order is a usable design parameter.** Definition 5.6 formalizes the largest $n$ for which an architecture accesses $\mathbb{R}[V^n]^G$; Table 1 places each architecture on this axis and connects it to task requirements (angles need $\nu\geq 2$; dihedrals $\nu\geq 3$; chirality needs parity-odd channels).
>
> 3. **The algebraic gaps survive finite-sample optimization.** Section 7 trains identical MLPs differing only in which invariants they consume: the $\mathrm{O}(3)$-restricted network reaches $R^2 < 0$ on a chirality task while the $\mathrm{SO}(3)$ network reaches $R^2=1$; a norm-only network cannot learn angles while a cross-invariant network learns them perfectly.
>
> 4. **Predictions for open questions in mainstream GDL.** The revision uses the framework to explain why PaiNN and MACE succeed on the symmetric counterexamples of Li, Wang, Kang et al. (ICLR'25) where Vanilla DisGNN fails — a question those authors explicitly left open — and to give algebraic accounts of the sparse-graph dichotomy of Sverdlov & Dym (ICLR'25) and the chirality failure of the Gaunt tensor product (Xie et al., ICML'25).
>
> **On pedagogy.** One way to read the reviewer's concern is that the operative word is *simply* — i.e., that the issue is invariant-ring vocabulary rather than absence of standard ML prose. On that reading, the concern is partially addressed: each architecture in §5.2 is introduced in standard ML terminology before the algebraic analysis (e.g., PaiNN as "augments scalars with vector features ($l=1$)," SchNet as "uses only scalar features with radial basis functions of pairwise distances"), with the algebraic interpretation following. The revision also restructures Section 5 so the architecture comparison leads rather than concludes, and adds §5.3 (Boundaries: Depth, Sparsity, Input Symmetry) in concrete architectural terms. We would welcome a pointer to which specific passages still read as inaccessible — §5.2's per-architecture bridges, the experimental setup in §7, or the algebraic preliminaries in §2 — so we can address the concern surgically rather than rewriting at a guess.
>
> **On audience.** The literature integrated in the revision — Cen et al. (NeurIPS'24, NeurIPS'25), Lin et al. (ICLR'26), Xie et al. (ICML'25), Sverdlov & Dym (ICLR'25), Li et al. (NeurIPS'23, ICLR'25), Joshi et al. (ICML'23), Delle Rose et al. (NeurIPS'23), Dym & Maron (ICLR'20) — is the mainstream geometric deep learning theory community at standard venues, and the framework gives unified algebraic explanations for results across these papers.

---

### Review · Reviewer_8ZCZ · 2026-04-28

**Summary Of Contributions:**

This paper discusses some drawbacks of the existence of isovariant networks and suggests that to overcome these drawbacks, more attention should be paid to aggregation methods rather than depth.

**Strength**

The theoretical explanation in this paper is clear and correct, and its approach from the perspective of algebraic structure is a relatively new one compared to previous articles, which I believe will inspire the community.

**Weakness**

The conclusion of this paper is quite obvious: models that only use distance (such as SchNet and EGNN) lack some information, while the product term can improve the network's expressive power—a view widely accepted in the field. Therefore, I believe there is still a deficiency in terms of contribution. This is not to emphasize novelty, but rather that I think the authors need to integrate more references using their algebraic systems-based perspective. See "Requested Changes" for details.

**Audience:**

Yes

**Audience Explanation:**

Although the conclusions drawn in this paper are well-known, and even somewhat trivial, I believe that analyzing equivariant neural networks from the perspective of algebraic structures is a relatively novel approach and will inspire new paradigms of thought within the community.

**Broader Impact Concerns:**

N/A.

**Claims And Evidence:**

Yes

**Claims Explanation:**

I read the theoretical proof in this article; it was simple and easy to understand, and I found no errors.

**Requested Changes:**

- Regarding the expressive power of product terms, I recommend the following references: Eq. (3) & Appendix C in [A], Sec. 6.2 in [B], Sec. 5.2 in [C].
- For further analysis of the model, I suggest introducing the latest spherical-scalarization, such as Eq. (10) & Thm. 4.1 in [D], Eqs. (10)-(11) in [E], as described in the analysis of PaiNN on page 8 of this paper.
- In addition, the approach of analyzing equivariant networks based on algebraic structure can be found in [F].
- Some articles discussing expressiveness of equivariant neural networks from other perspectives [G-J] could also be considered for mention.
- Finally, I suggest that authors label their formulas to facilitate citation by reviewers and future readers.

---
- [A] Cen J, Li A, Lin N, et al. Universally Invariant Learning in Equivariant GNNs. NeurIPS' 25.
- [B] Xie Y, Daigavane A, Kotak M, et al. The Price of Freedom: Exploring Expressivity and Runtime Tradeoffs in Equivariant Tensor Products. ICML'25.
- [C] Xie Y Q, Daigavane A, Kotak M, et al. Asymptotically Fast Clebsch-Gordan Tensor Products with Vector Spherical Harmonics[J]. arXiv:2602.21466, 2026.
- [D] Cen J, Li A, Lin N, et al. Are high-degree representations really unnecessary in equivariant graph neural networks? NeurIPS'24.
- [E] Aykent S, Xia T. Gotennet: Rethinking efficient 3d equivariant graph neural networks. ICLR'25.
- [F] Lin N, Cen J, Li A, et al. Reducing Symmetry Increase in Equivariant Neural Networks. ICLR'26.
- [G] Delle Rose V, Kozachinskiy A, Rojas C, et al. Three iterations of (d-1)-WL test distinguish non isometric clouds of d-dimensional points. NeurIPS'23.
- [H] Sverdlov Y, Dym N. On the expressive power of sparse geometric MPNNs. ICLR 2025.
- [I] Li Z, Wang X, Huang Y, et al. Is distance matrix enough for geometric deep learning? NeurIPS'23.
- [J] Li Z, Wang X, Kang S, et al. On the completeness of invariant geometric deep learning models. ICLR'25.

---

> ### Author Response · Authors · 2026-05-20
>
> We thank the reviewer for the careful read and substantial reference list. All ten papers [A]–[J] are now integrated, and several have reshaped how we frame the contribution.
>
> **On contribution.** We agree the central observation — that aggregation recovers expressivity that depth alone cannot — is widely accepted in the field. The revision repositions the contribution accordingly: not the existence of this gap, but an algebraic frame that gives a unified account for a body of recent constructive and algorithmic results. In the revised paper, the invariant-ring perspective is used to explain why $(d{-}1)$-WL needs three iterations [G], why the sparse-graph I-GGNN/E-GGNN dichotomy reduces to generic global rigidity vs. connectivity [H], why $k$-DisGNN scalar/vector universality splits at $k=2$/$k=3$ [I], why the symmetric case left open by [A] is structurally hard (stabilizer obstruction, with the trace criterion of [D] and full classification of [F] giving its algebraic content), why GeoNGNN-C's injected sign is exactly the generator of $\mathbb{R}[(\mathbb{R}^3)^n]^{\mathrm{SO}(3)}\setminus\mathbb{R}[(\mathbb{R}^3)^n]^{\mathrm{O}(3)}$ [J], and why the GTP chirality obstruction [B] is an operation-level missing selection rule restored by VSTP [C].
>
> **Beyond synthesis.** The algebraic frame yields predictions of its own. The clearest example: [J] explicitly leaves open the theoretical explanation of why PaiNN and MACE distinguish their symmetric counterexamples, alongside the four invariant architectures (DimeNet, SphereNet, GemNet, GeoNGNN) whose completeness they prove formally. Body-order analysis (§5.2) supplies that explanation: PaiNN reaches $\nu=2$ in a single layer via $\langle\vec v_i,\vec v_i\rangle$, accessing the cross-invariants $\hat r_{ij}\cdot\hat r_{ik}$ that resolve high-stabilizer configurations, and MACE's tensor products reach $\nu\geq 2$ by construction. The mechanism behind PaiNN's and MACE's empirical success is the same body-order $\geq 2$ access to $\mathbb{R}[(\mathbb{R}^3)^n]^{\mathrm{O}(3)}$ cross-invariants that makes DimeNet and GemNet succeed, implemented through different machinery. This closes a previously open theoretical question rather than restating a known one.
>
> **Requested references.** Each is now integrated as follows:
>
> - **[A] Cen et al. (NeurIPS'25)** — §5.1: their canonical form $\Gamma(\mathcal{G})$ via four-point positioning is framed as the constructive realization of aggregation for asymmetric configurations; the symmetric case they leave open is identified as the stabilizer-confined regime.
> - **[B] Xie et al., *Price of Freedom* (ICML'25)** — §5.2 and §8.2: their selection-rule analysis presented as the operation-level shadow of the invariant-ring picture, with GTP's exclusion of $1o\otimes 1o\to 1e$ giving the algebraic reason for chirality failure.
> - **[C] Xie et al., VSTP (arXiv:2602.21466)** — same locations: VSTP restores the parity-odd paths while keeping the asymptotic speedup.
> - **[D] Cen et al., HEGNN (NeurIPS'24)** — §5.2: now central. We extend the PaiNN analysis into a "spherical scalarization paradigm" hierarchy PaiNN ($l{=}1$) $\to$ HEGNN (all $l$) $\to$ GotenNet (attention over $l$), at fixed body-order $\nu=2$, and use HEGNN's trace criterion to explain degree collapse on $H$-symmetric inputs.
> - **[E] Aykent & Xia, GotenNet (ICLR'25)** — §5.2: most refined point on the parity-even $\nu=2$ axis.
> - **[F] Lin et al., ICLR'26** — §5.2 and §5.3: their full / axial / half degeneration spectrum and symmetry-infimum tabulation give a practical converse to body-order analysis, telling the practitioner which feature degrees are actively harmful for inputs with stabilizer $H$.
> - **[G] Delle Rose et al. (NeurIPS'23)** — §1.3, §5.1: their three-iteration $(d{-}1)$-WL completeness, especially 2-WL in $d=3$ and the 1-WL planarity obstruction, presented as the sharp algorithmic counterpart of Theorem 5.1.
> - **[H] Sverdlov & Dym (ICLR'25)** — §1.3, §5.3 (Boundaries: Sparsity): I-GGNN/E-GGNN dichotomy given an invariant-ring reading; scalar-only architectures inflate the aggregation radius to compensate for the lack of equivariant intermediate features.
> - **[I] Li, Wang, Huang et al., DisGNN (NeurIPS'23)** — §1.3, §5.1, §5.2: $k=2$ scalar / $k=3$ vector universality as the algorithmic shadow of the $\mathrm{O}(3)$ vs $\mathrm{SO}(3)$ ring distinction; DimeNet as 2-DisGNN and GemNet as 3-DisGNN as constructive justification of the body-order claims.
> - **[J] Li, Wang, Kang et al. (ICLR'25)** — §5.1, §5.2: measure-zero failure set characterization, node marking reframed as the message-passing realization of aggregation, and GeoNGNN-C's pseudoscalar injection as the algebraic prediction of SE(3)-completeness.
>
> **Equation labels.** All displayed equations are now labeled.

---

> ### Comment · Reviewer_8ZCZ · 2026-05-21
>
> The claim that "Body-order analysis (§5.2) supplies that explanation: ... This closes a previously open theoretical question rather than restating a known one" needs careful consideration. I understand the authors want to emphasize their contribution, but in fact, many related papers have made similar assertions, especially in the sources of MACE (Atomic Cluster Expansion method and related analysis). TMLR's acceptance criteria do not mandate novelty; therefore, I think the authors could more calmly explain the background of the field without getting bogged down in whether the problem has been solved before. Elaborately clarifying it again with elegant algebraic theory is also a valuable contribution.
>
> Furthermore, the fact that the models discussed in §5.2 are all "single-layer" needs to be emphasized. For some architectures, implicit construction of multi-body interaction structures can indeed be achieved by stacking layers.
>
> I believe that the above simple modifications would make this paper acceptable.

---

> > ### Author Response · Authors · 2026-05-29
> >
> > We thank the reviewer and have made both revisions.
> >
> > **On the PaiNN/MACE framing.** We agree. The §5.2 sentence has been rewritten more conservatively, and we do not claim priority over similar assertions, particularly in MACE-related sources.
> >
> > **On single-layer vs. stacking.** We added a note after the architecture table making explicit that the body-order $\nu$ in §5.2 is per single message-passing layer, with a cross-reference to the Remark in §5.3 that already quantifies how stacking raises effective body-order (the receptive-field mechanism standard in the MACE-related literature). To prevent any apparent tension with the depth theorem (§6), we added a brief reconciling sentence in both §5.3 and §6.

---

> > > ### Comment · Reviewer_8ZCZ · 2026-05-29
> > >
> > > I believe the current version meets the standards for acceptance by TMLR.

---

### Decision · Action_Editor_mbHD · 2026-06-12

**Recommendation:** Reject

**Audience:**

No

**Audience Explanation:**

This paper discusses the expressivity limits of equivariant neural networks. It focuses on the message aggregation mechanism in equivariant graph neural networks, arguing that aggregation enlarges the invariant ring of the mapping, increases the model’s ability to distinguish geometric relationships between nodes, and that many-body effects together with increased depth further improve performance. Although the paper offers a new perspective on the analysis of representational power, I recommend rejecting it because of vague definitions and factual errors.

**Claims And Evidence:**

No

**Claims Explanation:**

Even though two reviewers have recommended acceptance, there are some important issues serious enough to call its acceptance into question.

Strengths:

Beyond the compact groups commonly studied, such as O(3) and SO(3), the paper also considers non‑compact settings like GL(n) × GL(n), analysing them via FFT and PVS theory respectively. Starting from invariant theory, the paper does provide a fresh angle for analysing the representational capacity of equivariant networks.


Weaknesses:

1. Imprecise motivation

Section 1.2 (Motivation) contains factual errors and unclear descriptions. Such problems at the very core of the paper are, in my opinion, unacceptable for publication. The authors list three motivating points; I will address each in turn.

Scalar collapse: The claim that “two vectors pointing in different directions but with the same length produce proportional outputs” is incorrect. According to Malgrange’s theory, the correct statement should be that two vectors pointing in the same direction produce such behaviour.

Rank collapse: The statement “Networks equivariant to matrix transformations exhibit degeneracy at low rank” does not explain what “rank” refers to. I assume it means the rank of the map, but why equivariance leads to low rank is never properly justified.

Orbit limitations: This is the most problematic point. The authors write “More generally, equivariant networks cannot distinguish inputs that lie on the same group orbit.” Without any constraints on the inputs and outputs, this statement is false; it is, in fact, a property of invariant networks. There do exist situations where equivariant networks fail to distinguish inputs on the same orbit, when certain equivariant output features are chosen for symmetric inputs [1], but this is by no means general.

Overall, it appears the authors’ understanding of the representational power of equivariant networks is insufficient. They seem to be pointing out limitations that are inherent to the equivariance constraint. In reality, an equivariant map takes one orbit to another orbit, and the so‑called mode collapse only occurs at particular points or for particular choices of output equivariant features. This is not an inherent problem of equivariant mappings, but rather an issue of how input features are selected [1].

2. Limited contribution, insufficient clarity on compact groups, and missing applications for non‑compact groups.

Most application scenarios of equivariant networks involve compact groups (or groups that can be compactified via de‑centering), e.g., the SO(3) group in molecular and materials property prediction. Non‑compact groups may appear in high‑energy physics (the Lorentz group as a non‑compact Clifford group) [2]. For compact groups, the notion of Body‑Order introduced in Definition 5.6 is unclear: the role of the linear space V is ambiguous, and a rigorous connection to the form of the mapping is not established. For non‑compact groups, the authors consider the left‑ and right‑invertible matrix transformation groups, but no concrete application scenario is provided, nor are there any supporting numerical experiments.

3. Some conclusions are overly trivial, making the paper loose and unfocused.

Later in the text the authors repeat the slogan “invariant networks lose information.” This is obvious because an invariant network maps a whole group orbit to a single point, losing positional information, and therefore requires no example to support it. Similarly, Theorem 6.1 states that, in the context of invariant scalar outputs, “Points on the same G‑orbit cannot be separated, regardless of depth.” This is essentially the definition of invariance and does not merit being presented as a theorem with a proof.

4. Disorganised structure and missing appendices.

Material that should belong in an appendix is instead placed in the main text. For example, Section 5.1 contains an extensive discussion of related work that is not directly connected to the topic “Aggregation Escapes the Constraints.” This dilutes the main narrative and impedes readability; such content should be moved to an appendix.

The authors are suggested to submit a major revision to address the above issues.


References

[1] Joshi, Chaitanya K., et al. “On the expressive power of geometric graph neural networks.” International Conference on Machine Learning. PMLR, 2023.

[2] Ruhe, David, Johannes Brandstetter, and Patrick Forré. “Clifford group equivariant neural networks.” Advances in Neural Information Processing Systems 36 (2023): 62922–62990.

**Resubmission Of Major Revision:**

The authors may consider submitting a major revision at a later time.